# Structure Identification of ViceninII Extracted from *Dendrobium officinale* and the Reversal of TGF-β1-Induced Epithelial–Mesenchymal Transition in Lung Adenocarcinoma Cells through TGF-β/Smad and PI3K/Akt/mTOR Signaling Pathways

**DOI:** 10.3390/molecules24010144

**Published:** 2019-01-02

**Authors:** Yingyi Luo, Zhiyao Ren, Biaoyan Du, Shangping Xing, Shaowei Huang, Yunrong Li, Zhouxi Lei, Dan Li, Huanhuan Chen, Yuechun Huang, Gang Wei

**Affiliations:** 1School of Pharmaceutical Science, Guangzhou University of Chinese Medicine, Guangzhou Higher Education Mega Center, Guangzhou 510006, China; luofeiying@outlook.com (Y.L.); r._what@163.com (Z.R.); shopingxing@gmail.com (S.X.); 18150802287@163.com (S.H.); yyli0711025@163.com (Y.L.); leizhouxi@outlook.com (Z.L.); 13751717863@163.com (D.L.); chenhh1008@163.com (H.C.); huangyuechun@163.com (Y.H.); 2Department of Biochemistry, Guangzhou University of Chinese Medicine, Guangzhou 510006, China; dubiaoyan@gzucm.edu.cn; 3Shaoguan Institute of Danxia *Dendrobium Officinale*, Shaoguan 512005, China

**Keywords:** *Dendrobium officinale*, ViceninII, structural identification, EMT, TGF-β/Smad, PI3K/Akt/mTOR, lung adenocarcinoma, A549, H1299 cells

## Abstract

ViceninII is a naturally flavonoid glycoside extracted from *Dendrobium officinale*, a precious Chinese traditional herb, has been proven to be valuable for cancer treatment. Transforming growth factor-β1 (TGF-β1), promotes the induction of epithelial–mesenchymal transition (EMT), a process involved in the metastasis of cells that leads to enhanced migration and invasion. However, there is no previously evidence that ViceninII has an inhibitory effect on cancer metastasis, specifically on the TGF-β1-induced EMT process in lung adenocarcinoma cells. In this experiment, we used UV, ESIMS, and NMR to identify the structure of ViceninII.A549 and H1299 cells were treated with TGF-β1 in the absence and presence of ViceninII, and subsequent migration and invasion were measured by wound-healing and transwell assays. The protein localization and expressions were detected by immunofluorescence and Western blotting. The results indicated that TGF-β1 induced spindle-shaped changes, increased migration and invasion, and upregulated or downregulated the relative expression of EMT biomarkers. Meanwhile, these alterations were significantly inhibited when co-treated with ViceninII and inhibitors LY294002 and SB431542. In conclusion, ViceninII inhibited TGF-β1-induced EMT via the deactivation of TGF-β/Smad and PI3K/Akt/mTOR signaling pathways.This is the first time that the anti-metastatic effects of ViceninII have been demonstrated, and their molecular mechanisms provided.

## 1. Introduction

Lung cancer, the most frequent cancer in both sexes (18.4% of the total cancer cases in 2018), has the highest mortality rate of all malignant tumors globally [1,2]. Adenocarcinomas, which comprise approximately 80% of lung cancer cases [3], are classified as non-small-cell lung cancers (NSCLCs). Due to the lack of effective therapeutic strategies, the overall five-year survival rate of lung cancer patients is less than 15% when accompanied with metastasis [4], a process which causes about 90% of cancer-related deaths. Lung adenocarcinomas always metastasize at an early stage due to their high transfer ability [5]. Therefore, it is crucial to find an efficient compound for preventing metastasis in lung adenocarcinoma.

There is a growing body of evidence that the epithelial–mesenchymal transition (EMT) is closely associated with the migration, invasion, and subsequent metastasis of malignant tumors [6,7,8]. EMT is an essential process that occurs during development in which epithelial cells acquire mesenchymal, fibroblast-like properties and display reduced intracellular adhesion and increased motility [9,10,11]. Cadherins mediate calcium-dependent cell–cell adhesion and play critical roles in normal tissue development. The most critical hallmarks of the EMT are the downregulation of E-cadherin and the upregulation of N-cadherin through the loss of E-cadherin-mediated cell–cell junctions and tight junctions like ZO-1 and claudin-1, and in turn, the acquisition of mesenchymal markers including N-cadherin, vimentin, and fibronectin [12]. Matrix metalloproteinase-2 (MMP-2), a serine proteinase that can target growth factors, cell surface receptors, and adhesion molecules, has been reported to promote EMT signaling and stimulate tumorigenesis [13]. The EMT also involves the activation of transcription factors such as slug and snail, a widely expressed transcriptional repressor and member of the snail family of zinc finger transcription factors [14], which can bind to the E-cadherin promoter region to repress transcription during development [15]. 

Transforming growth factor (TGF)-β1, a potent pro-fibrotic factor, is a multifunctional cytokine and the most available stimulant of EMT. It promotes lung adenocarcinoma progression and metastasis [16,17,18]. Studies of the molecular signaling mechanism of TGF-β1-induced EMT indicate that the TGF-β/Smad signaling pathway plays a crucial role. TβRI and TβRII are two cognate receptors localized on the membrane. TGF-β1 binds to TβRII, leading to the activation and phosphorylation of TβRI [19]. Then, Smad2 and Smad3 are induced to undergo phosphorylation by the activated TβRI, forming heterodimeric complexes with Smad4. These complexes subsequently translocate into the nucleus, bind to the target genes, and regulate EMT through interactions with transcription factors such as slug and snail [20]. In addition, TGF-β1 can activate cell metastasis via signaling pathways uncorrelated with mad, including PI3K/Akt/mTOR [21,22], MAPK/ERK [23], NF-κB/P65 [12], and Rac1 [24]. Constitutive activation of PI3K/Akt/mTOR signaling cascades has been reported to play an essential role in the survival and metastasis of tumor cells [25]. When activated, the signal can be propagated through Akt, a downstream effector of PI3K, to mTOR, causing the phosphorylation of p70S6K, and finally leading to the rapid proliferation of tumor cells [26].

*Dendrobium officinale*, popularly called “Tiepi Shihu” in China is a perennial orchid plant listed individually in the China Pharmacopoeia (2010) among the 76 species [27]. It is a high-value medicinal herb and health-care food that has been widely used as a traditional Chinese medicine for thousands of years [28]. According to Chinese medicine theory, *(Bulpitt, 2007 #502; Yu, 2018 #503) D. officinale* clears heat and nourishes the lung, thereby treating lung-related diseases and improving the overall life quality of patients with lung cancer [29]. In modern pharmacological studies, *(Bulpitt, 2007 #502; Yu, 2018 #503) D. officinale* has displayed significant bio-active functions, including immunomodulatory [30,31], antioxidant [32,33], anti-inflammatory [34], antitumor [35,36] and hepatoprotective functions [37]. To date, most of the research in this area has been focused on polysaccharides and extracts from *(Bulpitt, 2007 #502; Yu, 2018 #503) D. officinale*, but flavonoids are a widespread group of significant substances with various biological pharmacological functions [38]. Unfortunately, only a small number of reports have focused on the pharmacological activities of flavonoids from *(Bulpitt, 2007 #502; Yu, 2018 #503) D. officinale* [39].

ViceninII, or 6,8-di-C-glucoside of apigenin, a nontoxic flavonoid that was originally identified from *Vitex lucens* wood and that is present in many plant sources in Southern Asia, such as *Dendrobium officinale*, *Urtica circularis*, *Lychnophora* and other species [40,41,42], is an effective anti-inflammatory [43,44,45,46], antiglycation [47], anti-spasmodic [48], antiseptic [49], antithrombotic, and antiplatelet agent [50]. ViceninII exhibits significant anticancer activities that inhibit the growth of prostate cancer (PC-3, DU-145 and LNCaP cells) in-vitro and in-vivo [51,52] and induce apoptosis in hT-29 human colon cancer cells [53]. In addition, many critical literatures has been reported that ViceninII has shown anti-inflammatory effect through the inhibition of TGF-β-induced protein signaling pathway as well as induced apoptosis of lung cancer H23 cell via PI3K/Akt/mTOR signaling [54,55]. However, none of the previous investigations investigated the anti-metastasis effects and its molecular mechanism of ViceninII in lung adenocarcinoma A549 and H1299 cells.

In this experiment, the structure of ViceninII was identified by UV, ESIMS, and NMR, and confirmed by comparing the spectral data following extraction from *Dendrobium officinale* leaves. The addition of 5 ng/mL of TGF-β1 can facilitate EMT and activate the TGF-β/Smad and PI3K/Akt/mTOR pathways in A549 and H1299 cells, so we assumed that ViceninII would reverse TGF-β1-induced EMT by inactivating these two signaling pathways. From these investigations, we present evidence that ViceninII prominently antagonizes TGF-β1-induced EMT by inactivating TGF-β/Smad and PI3K/Akt/mTOR pathways in lung adenocarcinoma A549 and H1299 cells. This is the first time that the anti-metastatic effect of ViceninII has been proven, and a reliable molecular mechanism provided. ViceninII may be a promising repressor against the metastasis of lung adenocarcinoma.

## 2. Results

### 2.1. Structural Identification of Vicenin*II*

The properties of ViceninII (**1**) are as follows: yellow amorphous powder; UV (MeOH) λ_max_ = 271 nm, 336 nm; ^1^H-NMR (MeOD, 500 MHz), δ: ppm 6.58 (1H, s, H-3), 7.94 (2H, d, *J* = 8.3 Hz, H-2′, H-6′), 6.89 (2H, d, *J* = 8.0 Hz, H-3′, H-5′), 4.92 (1H, d, *J* = 10.1 Hz, H-1″), 5.11(1H, d, *J* = 9,5 Hz, H-1′′′); ^13^C-NMR (MeOD, 125 MHz), δ: ppm 163.63 (C-2), 104.29 (C-3), 184.15 (C-4), 157.41 (C-5), 107.95 (C-6), 162.75 (C-7), 105.03 (C-8), 156.43 (C-9), 103.70 (C-10), 123.28 (C-1′), 130.06 (C-2′), 117.00 (C-3′), 161.67 (C-4′), 117.00 (C-5′), 129.68 (C-6′), 73.19 (C-1″), 70.95 (C-2″), 79.73 (C-3″), 71.11 (C-4″), 82.82 (C-5″), 61.77 (C-6″), 75.27 (C-1′′′), 71.66 (C-2′′′), 79.11 (C-3′′′), 70.95 (C-4′′′), 82.55 (C-5′′′), 63.15 (C-6′′′); ESI–MS *m*/*z* 593 [M−H]^−^, MS^2^
*m*/*z* 503 [(M−H)−C_3_H_6_O_3_]^−^, *m*/*z* 473 [(M−H)−C_4_H_8_O_4_]^−^, *m*/*z* 383 [(M−H)−C_3_H_6_O_3_−C_4_H_8_O_4_]^−^, *m*/*z* 353 [(M−H)−C_4_H_8_O_4_−C_4_H_8_O_4_]^−^; C_27_H_30_O_15._

Compound 1 turned fuchsia after reacting with magnesite powder and HCl, producing a purple cycle between the two liquid levels after the Molish reaction. UV spectrum absorption peaks were exhibited at 336 nm and 271 nm, suggesting the characteristics of the flavonoid glycoside skeleton. The ESI–MS results showed that the *m*/*z* of compound 1 was 593 [M−H]^−^ in the negative ion mode. It produced four prominent fragment ions at *m/z* 503 (C_24_H_23_O_12_), 473 (C_23_H_21_O_11_), 383 (C_20_H_15_O_8_), and 353 (C_19_H_13_O_7_) in the MS^2^ spectra. Therefore, the molecular weight of the compound was 594. The MS spectra, MS^2^ spectra, and molecular formula of the spectra of C_27_H_30_O_14_ are shown in Figure 1. 

The ^1^H-NMR spectrum of 1 showed a signal at δ 6.58 and the absence of H-6 and H-8, indicating that C-6 and C-8 had been substituted. The signals at δ 6.89 and δ 7.94 suggested that only a p-hydroxy group existed in ring B. The signals at δ 4.92 and δ 5.11 came from the anomeric positions of sugar. After the subtraction of 15 carbons from the flavone skeleton, another 12 carbons remained in the aglycone of 1. A comparison of the spectral data with the reference and literature data confirmed the identification of compound 1 as ViceninII [56]. The ^1^H-NMR and ^13^C-NMR spectra of ViceninII are shown in Appendix A.

### 2.2. The Cell Viability Effect and Morphology Changes of Vicenin*II* and Transforming Growth Factor 1 (TGF-β1) on Lung Adenocarcinoma A549 and H1299 Cells

In order to choose the appropriate concentration of TGF-β1 to induce EMT and ViceninII to avoid cell death in subsequent experiments, the cytotoxicity of ViceninII (1.25, 2.5, 5, 10, 20, 40, and 80 μM) and TGF-β1 (0.625, 1.25, 2.5, 5, 10, 20, and 40 ng/mL) on lung adenocarcinoma A549 and H1299 cells was measured by MTT [3-(4,5-dimethythiazol-2-yl)-2,5-diphenyl tetrazoliumbromide] assay after 24 h and 48 h. Following exposure to ViceninII (≤10 µM), A549 maintained greater than 95% viability and H1299 maintained 92% viability over 24 h, while higher doses of ViceninII (>10 µM) significantly suppressed A549 and H1299 cell viability in a dose- and time-dependent manner (Figure 2A). However, exposure to TGF-β1 at a concentration higher than 5 ng/mL exhibited an obvious cytotoxic effect by inhibiting the proliferation of A549 and H1299 cells over 48 h (Figure 2A, *p* < 0.05). Therefore, we selected concentration levels of ViceninII (2.5, 5 and 10 µM) and TGF-β1 (5 ng/mL) for further experiments.

As shown in Figure 2C, TGF-β1 (5 ng/mL) treatment successfully promoted the EMT by remarkably inducing the disappearance of intercellular junctions and spindle-like appearance of A549 and H1299 cells over 48 h, instead inducing a fibroblast-like appearance with a long shape and central nucleus. These morphological changes were obviously reversed by the cotreatment of TGF-β1 with ViceninII (2.5, 5, and 10 μM) for the last 24 h.

### 2.3. Clonogenic Formation in Human Lung Adenocarcinoma A549 and H1299 Cells

A colony formation assay is an in vitro cell survival assay based on the ability of a single cell to grow into a colony. To investigate the clonogenic potential and further anticancer activity of ViceninII (2.5, 5 and 10 μM) on its long-term efficacy against A549 and H1299 cells, a colony formation assay was performed. The results suggested that TGF-β1 treatment significantly increased colony amounts compared with the untreated cells, while significant dose-dependent decreases in the number of colonies were observed for ViceninII (10 μM) treatments in A549 and H1299 cells (Figure 3, *p* < 0.01).

### 2.4. Vicenin*II* Suppressed TGF-β1-Induced Migration and Invasion in A549 and H1299 Cells

The TGF-β1-mediated EMT of human lung cancer cells is closely related to the lung cancer metastatic characteristics, including migration and invasion. Compared with the control group, TGF-β1-treated cells showed highly enhanced migration potential in a wound-healing assay after 48 h, which was significantly suppressed by co-treatment with ViceninII (2.5, 5, and 10 µM) for the last 24 h (Figure 4A,B, *p* < 0.01). In addition, TGF-β1 obviously promoted invasion, as evidenced by the increased number of migrated cells in the transwell assay, which also dramatically decreased in a dose-dependent manner when co-treated with ViceninII (Figure 4A,B, *p* < 0.01). These results prove that ViceninII effectively inhibited the TGF-β1-induced migration and invasion of A549 and H1299 cells.

### 2.5. Vicenin*II* Reversed TGF-β1-Induced Expression of EMT Biomarkers in A549 and H1299 Cells

In order to find out whether ViceninII plays a critical role in TGF-β1-induced EMT in A549 and H1299 cells, we used Western blot and immunofluorescence assays to analyze the protein expression of EMT-associated biomarkers. The immunofluorescent staining results showed that TGF-β1-treatment obviously reduced the intensive green fluorescence expression of the epithelial marker E-cadherin and significantly increased the mesenchymal marker vimentin compared to control cells. These phenomena were completely reversed following co-incubation with 10 μM ViceninII (Figure 5A). In addition, the TGF-β1-induced protein expression of N-cadherin, vimentin, MMP-2, snail, and slug was significantly inhibited after treatment with ViceninII (2.5, 5, and 10 μM), whereas the expression of E-cadherin, Claudin-1, and ZO-1, which was inhibited by TGF-β1, increased obviously in a dose-dependent manner. These results demonstrate that ViceninII could effectively reverse TGF-β1-induced alterations in the protein expression of EMT biomarkers in A549 and H1299 cells.

### 2.6. The Inhibitory Effect of Vicenin*II* on TGF-β1-Induced EMT in Lung Cancer A549 and H1299 Cells through the TGF-β/Smad and PI3K/Akt/mTOR Pathways

TGF-β1 has been shown to activate the canonical Smad pathway and PI3K/Akt/mTOR pathways [16,22]. To analyze the signaling pathways involved in EMT reversion induced by ViceninII in A549 and H1299 cells, we detected the protein expression of the TGF-β1-induced TGF-β/Smad and PI3K /Akt /mTOR pathways. The band intensities shown in the western blot results (Figure 6A,B) indicate that the proteins p-Smad2, p-Smad3, p-mTOR, p-Akt and p-P70S6K were prominently up-regulated when the cells were treated with 5 ng/mL of TGF-β1, whereas total protein levels of Smad2, Smad3, mTOR, Akt, and P70S6K were essentially unchanged. However, when we added ViceninII (2.5, 5 and 10 µM) as a co-treatment, the phosphorylation of Smad2, Smad3, Akt, mTOR, and P70S6K decreased in a dose-dependent manner (Figure 6A,B, *p* < 0.05). 

Based on the above results, we further explored the functions of the TGF-β/Smad and PI3K/Akt/mTOR pathways by using the specific pharmacological inhibitors SB431542 (SB, TGF-β/Smad inhibitor) and LY294002 (LY, PI3K/Akt inhibitor). A549 and H1299 cells were incubated with 5 ng/mL of TGF-β1 for 48 h and subsequently, co-incubated with 10 µM ViceninII, 20 μM SB431542 and 20 µM LY294002 for the last 24 h. As previously demonstrated, down-regulation of E-cadherin and ZO-1, and up-regulation of N-cadherin, vimentin, MMP-2, snail, and slug by TGF-β1 were reverted by ViceninII in A549 and H1299 cells (Figure 6C, *p* < 0.05). Similar and even more effective results were observed following the LY and SB treatment, which was used as a positive control (Figure 6C, *p* < 0.05). These results demonstrate, once again, that ViceninII antagonizes TGF-β1-induced EMT through inhibition of the TGF-β/Smad and PI3K/Akt/mTOR pathways in lung adenocarcinoma A549 and H1299 cells.

## 3. Discussion

The aim of this study was to analyze the anti-metastatic effects of ViceninII on TGF-β1-induced EMT in lung adenocarcinoma cell lines and to determine its possible molecular mechanism(s) of action. 

Lung cancer has the highest morbidity and frequency of all cancers worldwide [1]. Metastasis of lung cancer cells to other vital organs initially happens at the early stages of the disease and leads to an overall five year survival rate of less than 15% [4,57]. Thus, the suppression of lung adenocarcinoma metastasis is vital to improve the survival of lung adenocarcinoma patients. Growing evidence suggests that the EMT is involved in the metastasis process [7], and many studies have reported that TGF-β1 contributes to the induction of EMT through the regulation of E-cadherin, N-cadherin, vimentin, snail, and MMP-2. Other representative characteristics of EMT also account for the migration and invasion of lung cancer cells [58,59]. However, Zheng et al. reported that in genetically engineered mouse models of pancreatic adenocarcinoma development, carcinoma cells could metastasize without activating EMT programs [60]. But the conclusion that EMT is not required for metastatic dissemination would require further evidence [61]. Hence, it is extremely important to develop a novel anti-metastasis agent that targets EMT-related pathways to benefit patients with metastatic lung adenocarcinoma. Due to Chinese medicine theory claims that it clears heat and nourishes the lungs, Dendrobium officinale is widely used in China to treat lung cancer as an adjunctive therapy in order to improve the life quality of patients. ViceninII, a C-glycoside flavonoid extracted from Dendrobium officinale, has been identified and isolated from various medicinal herbs and has shown multiple pharmacological activities, particularly anti-cancer activities [51,52,53], but none of the previous research has focused on the anti-metastasis effects. Therefore, we wondered whether ViceninII could inhibit the TGF-β1-mediated EMT metastasis of lung cancer cells.

In this study, an MTT assay showed that ViceninII has no toxic effect at concentrations less than 10 μM. If the does of ViceninII exceed 10 μM, it would has the killer effect on cells and we could not guarantee the anti-metastatic effect wasn’t due to its cytotoxic effects. Treatment with TGF-β1 at >10 ng/mL has an obvious cytotoxic effect as anastomosed in the reports on the regulation of cell proliferation and apoptosis [62]. Meanwhile, some literature has reported the successful induction of EMT on A549 and H1299 cells with 5 ng/mL of TGF-β1 [12,16], and the results of morphology changes prove this. Therefore, we finally chose concentration levels of ViceninII (2.5, 5 and 10 µM) and TGF-β1 (5 ng/mL) to proceed with in further experiments. Furthermore, we found that ViceninII obviously inhibits TGF-β1-induced cloning efficiency, migration, and invasion in a dose-dependent manner, as evidenced by the responses of A549 and H1299 cells to the clonogenic formation, wound healing, and transwell assay.

EMT is characterized by morphological changes of cells, the downregulation of epithelial markers (E-cadherin, ZO-1 and Claudin-1), and the upregulation of mesenchymal markers (N-cadherin and vimentin) and transcription factors (snail and slug) [9]. Members of the matrix metalloproteinase (MMP) family (such as MMP-2) are required for tumor invasion and metastasis [13]. Immunofluorescence staining and western blot analyses demonstrated that ViceninII remarkably reverses the aforementioned EMT-related protein expression induced by TGF-β1. Subsequently, we investigated the mechanisms through which this occurs. The association of TGF-β/Smad and PI3K/Akt/mTOR signaling pathways with the molecular mechanisms of EMT in tumor cells is receiving increasing attention in research [63,64,65]. In the current study, TGF-β1 upregulated the phosphorylate levels of Smad2/3, AKT, and mTOR in parallel with p70S6K, but these effects were inhibited in a dose-dependent manner after the addition of ViceninII, which preliminarily revealed that ViceninII works through the TGF-β/Smad and PI3K/Akt/mTOR pathways. 

LY294002, PI3K/Akt pathway inhibitor, has been shown to block PI3 kinase-dependent Akt phosphorylation and kinase activity [66]. SB431542, TGF-β/Smad pathway inhibitor, is a potent and selective ATP-competitive inhibitor of the TGF-β1 activin receptor-like kinases [67]. To further explore the relationship between the EMT and the TGF-β/Smad and Akt/mTOR/P70S6K pathways in lung adenocarcinoma, we used the LY294002 and the SB431542 to confirm that ViceninII plays a suppressive role in the EMT by regulating these pathways, as evidenced by the expression of E-cadherin, N-cadherin, ZO-1, vimentin, snail, slug, and MMP-2. The selection of inhibitor does (20 μM) was referred to the previous literature and specification [39]. The results of the group added inhibitors are similar to those found when co-treating with ViceninII. That is to say, as depicted in Figure 7, ViceninII reverses TGF-β1-mediated EMT via deactivating the TGF-β/Smad and PI3K/Akt/mTOR signaling pathways in A549 and H1299 cells.

Previous studies have reported that isoviolanthin, a C-glycoside flavonoid with apigenin as the parent nucleus, inhibits TGF-β1-induced EMT in hepatocellular carcinoma cells via the deactivation of the TGF-β/Smad and PI3K/Akt/mTOR signaling pathways [39]. ViceninII, known as apigenin-6,8-di-C-β-d-glucoside, belongs to the rare C-glycoside group of flavonoid glycosides consisting of a flavonoid moiety and two sugar units [38]. The nuclear parent of ViceninII is apigenin and the sugar units, located in C-6 and C-8 respectively, are both glucose compounds. The structure of Vicenin-2 is very similar to that of isoviolanthin. Both of them have an apigenin moiety and C-8 glucose. The difference between them is that the isoviolanthin has a rhamnose unit in C-6. Thus, we speculate that the structural characteristic of the apigenin moiety and polyhydroxyl group may affect the TGF-β/Smad and PI3K/Akt/mTOR signaling pathways. Furthermore, we have identified other compounds with this structural characteristic, and the signaling pathways of those compounds will be experimentally verified in the future.

In summary, the study provides strong evidence that ViceninII inhibits TGF-β1-induced EMT migration and invasion in lung adenocarcinoma A549 and H1299 cells. This is the first time these novel insights into the anti-metastatic effect of ViceninII as a potential repressor have been demonstrated.

## 4. Materials and Methods 

### 4.1. Plant Material and Reagents

Leaves of *Dendrobium officinale* were obtained from the Xinyu Ecological Development Co., Ltd. (Shaoguan, Guangdong, China), and authenticated by Professor Gang Wei (Pharmaceutical Department, Guangzhou University of Chinese Medicine). MeOD was purchased from Sigma (St. Louis, MO, USA). Petroleum ether, ethanol, n-butyl alcohol, and other reagents were of analytical grade.

### 4.2. Extraction and Isolation 

The dried Dendrobium officinale leaves (2 kg) were ground and extracted in 70% ethanol (3 × 5 L × 2 h) at room temperature. The extractions were filtered and concentrated to a residue (1183 g), which was suspended in an adequate amount of distilled water. Two volumes of petroleum ether, ethyl acetate, and n-butyl alcohol were sequentially added into the suspension for the extraction. A portion of the n-butyl alcohol extract (142 g) was chromatographed on AB-8 macroporous absorption resin (Guangzhou Wanye Chemical CO., Ltd., Guangzhou, China) and eluted with ethanol–H_2_O (0→15%→30%→50%→90%) to obtain four fractions. The 30% fraction was subjected to ODS (Tianjin Bohong Resin Technology CO., Ltd., Tianjin, China), and eluted with methanol–H_2_O (0→20%→30%→40%→50%). The 30% fraction was further purified by recrystallization to obtain compound **1**. 

### 4.3. Structural Analysis of Vicenin*II*

The characterization of the flavonoid was done with magnesite powder and HCl. The sugar substituent was evaluated by the Molish reaction. UV, ESIMS (Thermo Finnigan LCQ FLEET, Riviera Beach, FL, USA), ^1^H-NMR and a ^13^C-NMR (Avance III HD 500 MHz Digital NMR spectrometer Bruker, Karlsruhe, Germany) were used to further analyze the structure.

### 4.4. Cell Culture and Drug Treatment

Human lung adenocarcinoma cell line A549 and H1299 cells were gifted from Dr. Biaoyan Du, Professor, Guangzhou University of Chinese Medicine, Guangzhou, China. The A549 and H1299 cells were cultured in high-glucose Dulbecco’s Modified Eagle’s Medium (H-DMEM), supplemented with 10% fetal bovine serum (FBS), 0.5% streptomycin, and penicillin, at 37 °C in a humidified atmosphere of 5% CO_2_. All media components were obtained from Gibco (Grand Island, NY, USA). The concentration of TGF-β1 (PeproTech, Rocky Hill, NJ, USA) used to induce EMT was chosen by the MTT assay and morphology observation, based on what some literature has reported previously [58]. In order to guarantee the final DMSO content was less than 0.1%, the cell culture solution of ViceninII was prepared by first dissolving freeze-dried powder with DMSO (Sigma, St. Louis, MO) at a concentration of 10 mM, and then diluting it with H-DMEM medium to different concentrations (2.5, 5, and 10 μM).

### 4.5. Effect of Vicenin*II* and TGF-β1 on Cell Viability

The cell viability was estimated using MTT assays. A549 and H1299 cells were seeded in a 96-well plate (3 × 10^3^/well) and cultured for 24 h. Then, the culture medium was replaced by various concentrations of ViceninII (1.25, 2.5, 5, 10, 20, 40, and 80 μM) and TGF-β1 (0.625, 1.25, 2.5, 5, 10, 20, and 40 ng/mL). After incubation for 24 and 48 h, 20 µl MTT (5 mg/mL, Sigma-Aldrich, St. Louis, MO, USA) was added to each well and sequentially incubated for an additional 4 h at 37 ℃. The cell supernatant was removed and 100 µL DMSO was added to dissolve the blue formazan. The absorbance was measured at 490 nm using a micro-plate reader (Bio-Rad Laboratories, Inc., Hercules, CA, USA) after the plate was shaken for 5 min. The cell viability was calculated as follows:

Cell viability = (A490 experimental/A490 control) × 100%.

### 4.6. Morphology Observations

A549 and H1299 cells (1.5 × 10^5^/well) were seeded in 6-well plates, incubated overnight, and treated with TGF-β1 (5 ng/mL) alone or in combination with ViceninII (2.5, 5 and 10 μM) for 48 h. Cell morphology was captured by a bright-field microscope (Olympus, Hamburg, Germany).

### 4.7. Colony Forming Assay

A549 (1 × 10^3^ cells/well) and H1299 (8 × 10^2^ cells/ well) cells were plated as single cells onto a 6-well Petri dish overnight. Then, they were co-treated with ViceninII (2.5, 5, and 10 μM) and TGF-β1 (5 ng/mL) for 14 days. Colonies were fixed with 4% paraformaldehyde and stained with 0.1% crystal violet. Subsequently, cell plates were photographed and counted by Image-Pro Plus 6.0 software (Bethesda, Rockville, MD, USA). The result represent three separate experiments.

### 4.8. Wound Healing Migration Assay

A549 and H1299 cells (3 × 10^5^ cells/well) were seeded in 6-well plates for 24 h and a clear cell-free line was manually created by scratching the confluent monolayers with a 100 μL pipette tip in each well. The wounded monolayers were washed three times with PBS and photographed using an inverted microscope (Olympus, Hamburg, Germany) as the picture for 0 h. Then, the cells were co-treated with TGF-β1 (5 ng/mL) for 48 h in combination with ViceninII (2.5, 5 and 10 μM) for the last 24 h. TCell migration was captured as images at 48 h. The migration area was counted by Image-Pro Plus 6.0 software (Bethesda, MD, USA).

### 4.9. Transwell Invasion Assay

A549 and H1299 cells (2 × 10^4^ cells/well) were plated in Matrigel-coated top chambers (8 μm pore size, Corning, New York, NY, USA) with 200 µL of serum-free H-DMEM containing ViceninII (2.5, 5 and 10 μM) for 24 h. Meanwhile, 800 µL of H-DMEM (10% FBS) medium with TGF-β1 (5 ng/mL) was used as a chemoattractant in the lower chambers. After incubation for 24 h, the non-migrated cells were removed from the upper surface of the chamber, and the invaded cells were fixed with 4% paraformaldehyde for 10 min and stained with 0.1% crystal violet for 15 min. Then, the invading cells were photographed and counted by Image-Pro Plus 6.0 software (Bethesda, MD, USA).

### 4.10. Immunofluorescence Assay

A549 and H1299 cells (2 × 10^4^ cells/well) were seeded in glass slides on 12-well plates and incubated for 24 h. The cells were treated with TGF-β1 (5 ng/mL) for 48 h and ViceninII (10 μM) for the last 24 h. The cell slides were removed from the medium and fixed in 4% paraformaldehyde at room temperature for 15 min, blocked with 5% bovine serum albumin (BSA) and 0.5% Triton X-100 for 2 h at 37 °C, and then incubated overnight at 4 °C with anti-vimentin and anti-ecadherin antibodies (Cell Signaling Technology, Danvers, MA, USA, 1:100). After this, the cell slides were incubated with secondary fluorescein isothiocyanate (FITC)-conjugated anti-rabbit IgG antibody (Cell Signaling Technology, 1:1000) for 1 h and counterstained with 4′,6-diamidino-2-phenylindole (DAPI) (Beyotime, Guangzhou, China) for 20 min at room temperature. The cells were viewed and photographed under laser confocal microscopy 880 with Airyscan (Zeiss, Jena, Germany). 

### 4.11. Western Blot Analysis

A549 and H1299 cells (1.5 × 105 cells/well) were plated in 6-well plates for 24 h, followed by incubation with TGF-β1 (5 ng/mL) for 48 h and ViceninII (2.5, 5 and 10 μM) or PI3K/Akt inhibitor LY294002 (20 μM, MedChem Express, NJ, USA) and TGF-β/Smad inhibitor SB431542 (20 μM, MedChem Express) for the last 24 h. Total protein was extracted from the cells using RIPA buffer (radioimmunoprecipitation assay buffer) added to 1% PMSF (phenylmethanesulfony fluoride). A BCA protein quantitative method was used to quantify the concentration of each protein sample. Then, 20 μL protein samples were separated by 8–12% SDS-PAGE and transferred to PVDF membranes (Millipore, Bedford, MA, USA). The membranes were cut according to the molecular weight of the protein of interest by comparison with the protein marker. Then, 5% nonfat dry milk protein dissolved in TBS-T (0.1% Tween-20) was used to block each membrane at room temperature for 3~4 h, followed by incubation with the antibodies E-cadherin, N-cadherin, ZO-1, β-actin, vimentin, MMP-2, snail, slug, Claudin-1, p-mTOR, mTOR, P70S6K, p-P70S6K (1:1000 dilution, Cell Signaling Technologies, Beverly, MA, USA) and p-Smad2, Smad2, p-Smad3, Smad3, p-Akt, and Akt (1:1000 dilution, ABclonal, Wuhan, China) at 4 °C for 12~18 h. Next, each membrane was incubated with the corresponding secondary horseradish peroxidase (HRP)-conjugated goat anti-rabbit or anti-mouse IgG antibodies (1:4000 dilution, ABclonal, Wuhan, China) at room temperature for 2 h. In the end, the protein expression was developed using enhanced chemiluminescence HRP substrate (Millipore, Bedford, MA, USA) and detected by the Tanon detection system (Shanghai, China). The gray value of each protein band was scanned by Image-Pro Plus 6.0 software (Bethesda, MD, USA).

### 4.12. Statistical Analysis

Data are shown as the mean ± standard deviation (SD) and the results from each group were obtained using triplicate samples independently. The MS data were analyzed with the Finnigan Xcalibur 2.0 advanced chromatography workstation (Thermo Quest Corporation, San Jose, CA, USA). A one-way ANOVA was used to determine the significant differences among the groups. A probability value of *p* < 0.05 was considered to be statistically significant. All statistical analyses were conducted with SPSS 20.0 (SPSS Inc., Chicago, IL, USA) and Prism 6.0 software (GraphPad Software Inc., La Jolla, CA, USA).

## 5. Conclusions

ViceninII will be one of the most important components responsible for the anti-metastasis effect of Dendrobium officinale. For the first time, we demonstrate that ViceninII targets the TGF-β/Smad and PI3K/Akt/mTOR pathways to inhibit TGF-β1-induced EMT phenotypes in lung adenocarcinoma A549 and H1299 cells. Furthermore, these results provide evidence that ViceninII could be a promising repressor against the metastasis of lung adenocarcinoma by affecting TGF-β1-induced EMT.

## Figures and Tables

**Figure 1 molecules-24-00144-f001:**
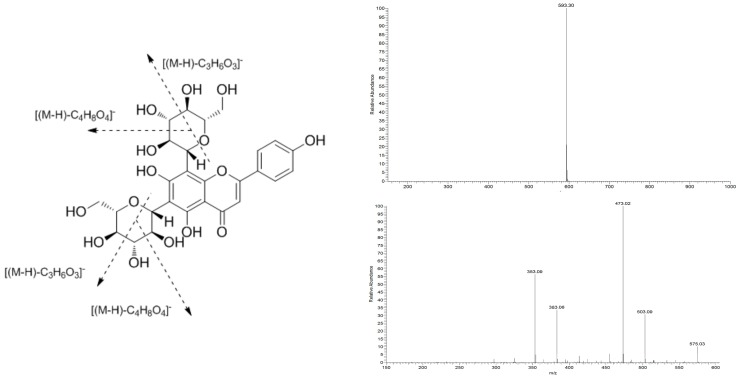
Structure and MS/MS spectra of ViceninII in negative ion mode.

**Figure 2 molecules-24-00144-f002:**
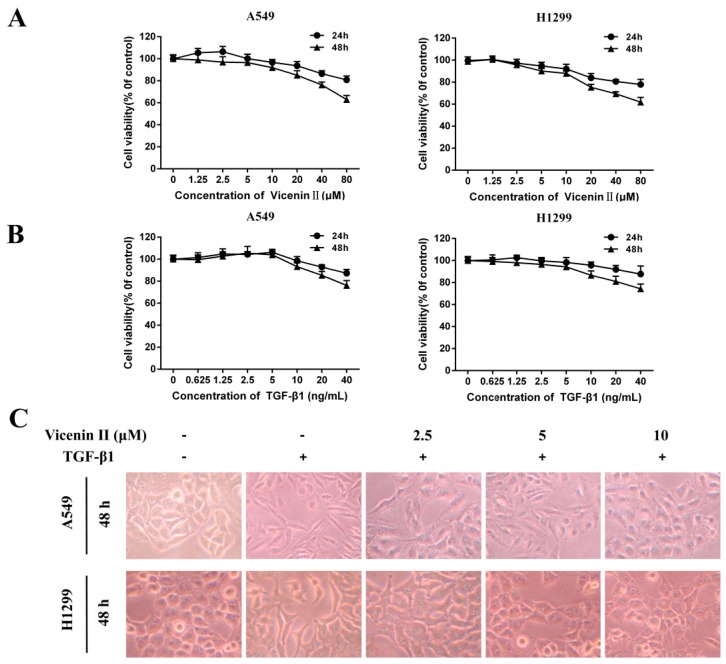
Effects of ViceninII and transforming growth factor (TGF)-β1 on the viability and morphology changes of lung adenocarcinoma A549 and H1299 cells. The cell viability of A549 and H1299 cells treated with (**A**) ViceninII (1.25, 2.5, 5, 10, 20, 40 and 80 μM), (**B**) TGF-β1 (0.625, 1.25, 2.5, 5, 10, 20 and 40 ng/mL) for 24 h and 48 h was determined by MTT [3-(4,5-dimethythiazol-2-yl)-2,5-diphenyl tetrazoliumbromide] assay. Cells were treated with TGF-β1 (5 ng/mL) for 48 h and co-treated with ViceninII for the last 24 h, and the cell morphology was observed (**C**). Magnification ×200.

**Figure 3 molecules-24-00144-f003:**
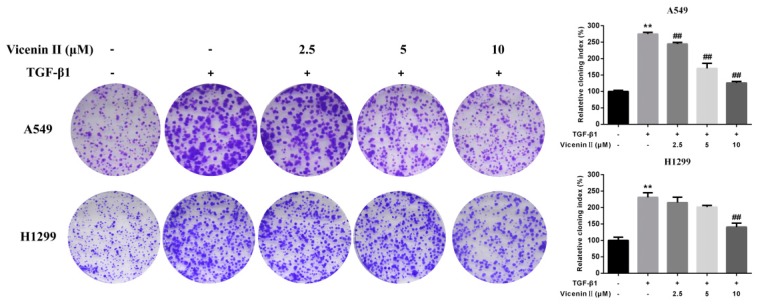
Effects of ViceninII on colony formation in human lung A549 and H1299 cells. After being co-incubated with 5 ng/mL of TGF-β1 and ViceninII (2.5, 5 and 10 μM) for 14 days, the clonogenic potential of each group was analyzed by colony formation assay. Representative graphs and a histogram (mean ± SD) are shown, *n* = 3. ** *p* < 0.01 compared with the control group; ^##^
*p* < 0.01 compared with the TGF-β1 group.

**Figure 4 molecules-24-00144-f004:**
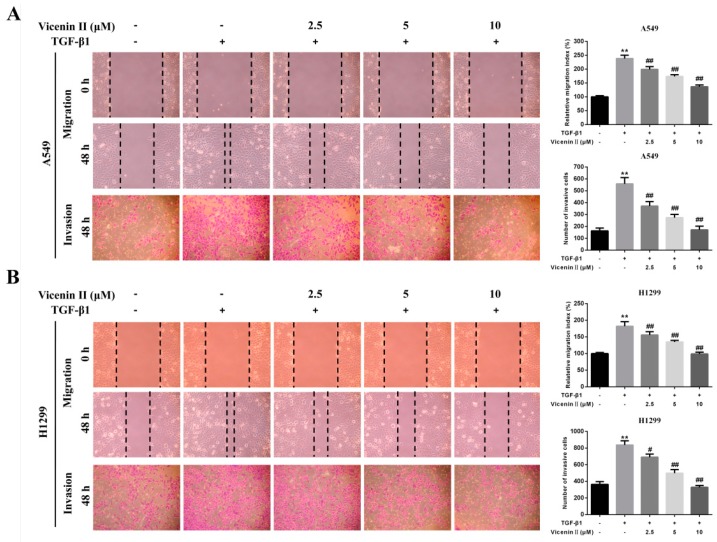
ViceninII inhibited TGF-β1-induced migration and invasion of (**A**) A549 and (**B**) H1299 cells. Cells were treated with TGF-β1 (5 ng/mL) for 48 h and co-treated with ViceninII (2.5, 5 and 10 µM) for the last 24 h. The migration and invasion capacities were measured by the wound healing and transwell assays. Typical graphs and a histogram (mean ± SD) are shown, *n* = 3. Magnification is ×100. ** *p* < 0.01 compared with the control group; ^#^
*p* < 0.05, ^##^
*p* < 0.01 compared with the TGF-β1-treated group.

**Figure 5 molecules-24-00144-f005:**
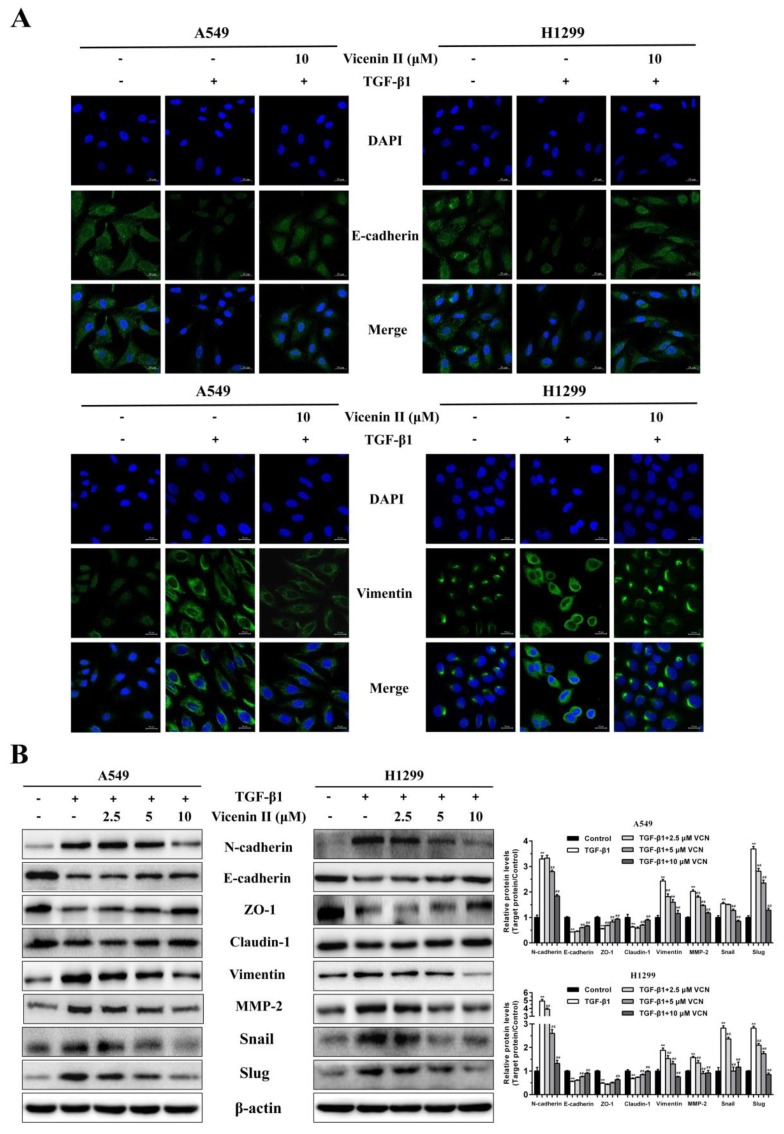
Effect of ViceninII on the expression of TGF-β1-induced EMT biomarkers in A549 and H1299 cells. (**A**) After treatment with TGF-β1 (5 ng/mL) for 48 h and co-treatment with 10 µM ViceninII for the last 24 h, immunofluorescence staining was performed to detect E-cadherin and vimentin expression in A549 and H1299 cells by confocal microscopy. Green fluorescence indicates E-cadherin and vimentin positive expression, and blue fluorescence indicates 4′,6-diamidino-2- phenylindole (DAPI)-labeled nuclei. (**B**) Cells were treated with TGF-β1 (5 ng/mL) for 48 h and co-treated with ViceninII (2.5, 5 and 10 µM) for the last 24 h, and the protein expression of EMT-associated biomarkers was determined by western blot analysis. Scale bars: 20 μm. Typical graphs and a histogram (mean ± SD) are shown, *n* = 3. ** *p* < 0.01 compared with the control group; ^##^
*p* < 0.01 compared with the TGF-β1-treated group.

**Figure 6 molecules-24-00144-f006:**
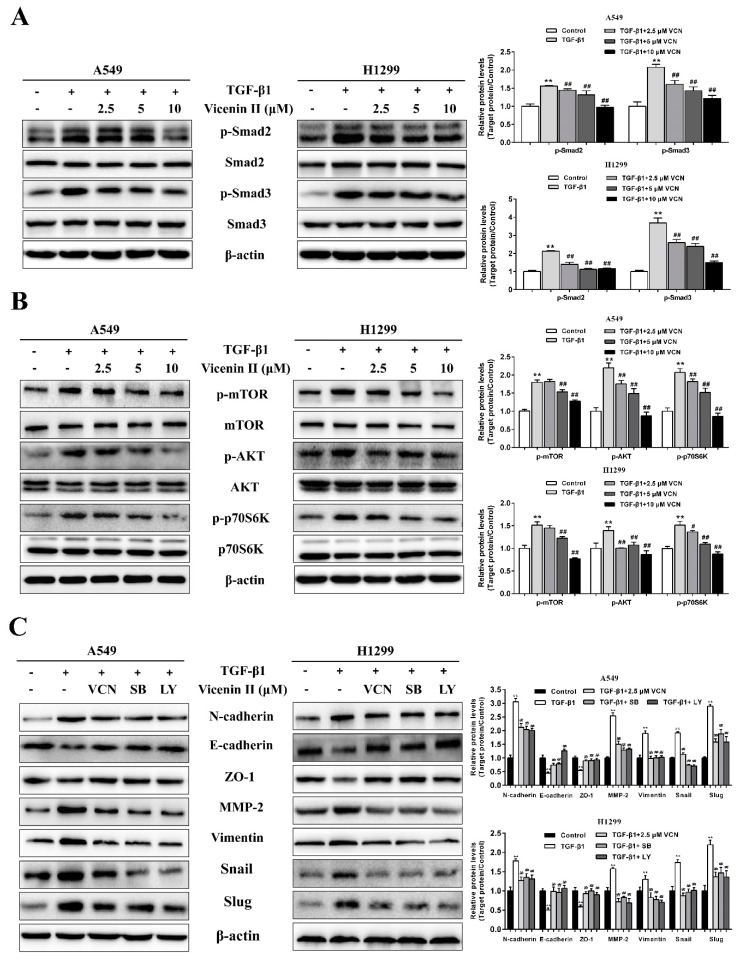
Effects of ViceninII on the TGF-β/Smad and PI3K/Akt/mTOR pathways in A549 and H1299 cells. (**A**) Cells were treated with 5 ng/mL of TGF-β1 for 48 h, during which the indicated concentrations of ViceninII were added for the last 24 h. The expression of TGF-β/Smad related pathway proteins, including p-Smad2, Smad2, p-Smad3, and Smad3, was examined by western blot analysis. (**B**) The expression of PI3K/Akt/mTOR related pathway proteins, such as p-Akt, Akt, p-mTOR, mTOR, p-P70S6K and P70S6K, was detected by western blot analysis. It was reconfirmed that ViceninII suppresses TGF-β1-induced EMT in A549 and H1299 cells through regulation of the TGF-β/Smad and PI3K/Akt/mTOR pathways. (**C**) Cells were pre-treated with 5 ng/mL TGF-β1 for 48 h and then treated with 10 μM ViceninII, 20 μM SB431542 (SB, TGF-β/Smad inhibitor) or 20 µM LY294002 (LY, PI3K/Akt inhibitor) for the last 24 h. Then, using western blot, the protein expression of E-cadherin, ZO-1, N-cadherin, MMP-2, vimentin, snail, and slug was analyzed. β-actin was used as a loading control. Typical graphs and a histogram (mean ± SD) are shown, *n* = 3. ** *p* < 0.01 compared with the control group; ^##^
*p* < 0.01 compared with the TGF-β1-treated group.

**Figure 7 molecules-24-00144-f007:**
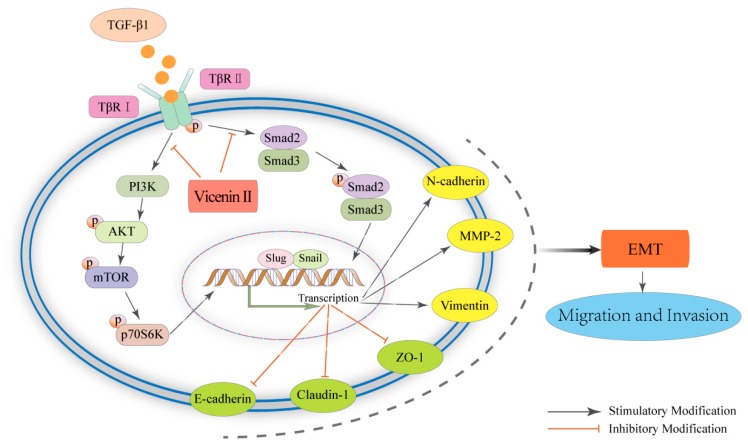
Schematic representation of the effect of ViceninII on TGF-β1-induced EMT in lung adenocarcinoma A549 and H1299 cells. ViceninII suppressed the PI3K/Akt/mTOR and TGF-β/Smad pathways and finally inhibited migration and invasion of lung adenocarcinoma cells by affecting TGF-β1-induced EMT. “→” indicates promotion; “⊥” indicates inhibition.

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
