# Peer review of "Structure Identification of ViceninII Extracted from Dendrobium officinale and the Reversal of TGF-β1-Induced Epithelial–Mesenchymal Transition in Lung Adenocarcinoma Cells through TGF-β/Smad and PI3K/Akt/mTOR Signaling Pathways"

_molecules, 2019, doi:10.3390/molecules24010144_

Round 1

Reviewer 1 Report

In this manuscript, Luo et al have investigated the potential effect of a herbal derivative on development of EMT in lung cancer cells. The study has been well designed to address a relevant important question. However, there are following concerns to be resolved before acceptance.

1) Specificity to TGF. It would make more sense to study whether the effect of Vicenin II on EMT of lung tumor cells  is mediated exclusively through TGF or not. 

2) Type of cells. The authors should discuss the rationale behind choosing two different cell types in terms of mutations and other characteristic features. 

3) Contraversial  findings on the role of EMT in metastasis should be critically discussed. 

4) The effect of Vicenin II at the baseline without TGF stimulation needs to be included in both functional assays and mechanistic studies as a necessary control group. 

5) Pathophysiological relevance of TGF, inhibitors, and Vicenin II doses examined should be extensively explained in the Discussion. 

6) Grammatical errors should be professionally corrected.

Author Response

    In this manuscript, Luo et al have investigated the potential effect of a herbal derivative on development of EMT in lung cancer cells. The study has been well designed to address a relevant important question. However, there are following concerns to be resolved before acceptance.

Point 1: Specificity to TGF. It would make more sense to study whether the effect of Vicenin II on EMT of lung tumor cells  is mediated exclusively through TGF or not.

Response 1: Thank you for your valuable suggestion.We use TGF-β1 to induce metastasis in this investigation and we mainly focus on the anti-metastatic effect of Vicenin II in TGF-β1-induced lung cancer cells.We want to find out what alterations Vicenin II was responsible for some of the observed results induced by TGF-β1 and evaluate the molecular mechanism.We are interested to study whether the effect of Vicenin II on EMT of lung tumor cells is mediated exclusively through TGF or not in the future.

Point 2:Type of cells. The authors should discuss the rationale behind choosing two different cell types in terms of mutations and other characteristic features. 

Response 2: Thank you for your nice comments on our manuscript.Some studies also suggested that lung cancer cells A549 and H1299 are suitable models for the study of EMT induced by TGF-β1 in vitro.

For example:

Feng H,Lu JJ,Wang Y,et al.Osthole inhibited TGF β-induced epithelial-mesenchymal transition (EMT) by suppressing NF-κB mediated Snail activation in lung cancer A549 cells[J].Cell adhesion & migration,2017,11(5-6):464-475.DOI:10.1080/19336918.2016. 1259058

[2] Da C,Liu Y,Zhan Y,et al.Nobiletin inhibits epithelial-mesenchymal transition of human non-small cell lung cancer cells by antagonizing the TGF-β1/Smad3 signaling pathway[J].Oncology reports,2016,35(5):2767-2774.DOI:10.3892/or.2016.4661

[3] Lin CY,Hsieh YH,Yang SF,et al.Cinnamomum cassia extracts reverses TGF-β1-induced epithelial-mesenchymal transition in human lung adenocarcinoma cells and suppresses tumor growth in vivo[J].Environmental toxicology,2017,32(7):1878-1887.DOI:10.1002/tox.22410

We haved observed the characteristic features of A549 and H1299 cells by a microscope and described it’s morphology changes in the results (lines 150-154).

Point 3:Contraversial findings on the role of EMT in metastasis should be critically discussed. 

Response 3: Thank you for again for your positive comments to improve the quality of our manuscript.We have added the contraversial findings on the role of EMT in metastasis in Discussion (lines 264-267).

Point 4:The effect of Vicenin II at the baseline without TGF stimulation needs to be included in both functional assays and mechanistic studies as a necessary control group. 

Response 4:Thank you for your suggestion. In our investigation, we mainly focus on the anti-metastatic effect of Vicenin II in TGF-β1-induced lung cancer cells.We want to find out what alterations Vicenin II was responsible for some of the observed results induced by TGF-β1 and evaluate the molecular mechanism.We feel sorry that we can’t provide another expriment group without TGF-β1 in this manuscript. However, as far as we know, numerous excellent studies was similar to our investigation that didn’t added the group without TGF-β1.As follows:

[1] Feng H,Lu JJ,Wang Y,et al.Osthole inhibited TGF β-induced epithelial-mesenchymal transition (EMT) by suppressing NF-κB mediated Snail activation in lung cancer A549 cells[J].Cell adhesion & migration,2017,11(5-6):464-475.DOI:10.1080/19336918.2016.1259058

[2] Da C,Liu Y,Zhan Y,et al.Nobiletin inhibits epithelial-mesenchymal transition of human non-small cell lung cancer cells by antagonizing the TGF-β1/Smad3 signaling pathway[J].Oncology reports,2016,35(5):2767-2774.DOI:10.3892/or.2016.4661

[3] Kim YJ,Choi WI,Jeon BN,et al.Stereospecific effects of ginsenoside 20-Rg3 inhibits TGF-β1-induced epithelial-mesenchymal transition and suppresses lung cancer migration, invasion and anoikis resistance[J].Toxicology,2014,322():23-33.DOI:10.1016/j.tox.2014.04.002

[4] Zhai XX,Ding JC.Resveratrol Inhibits Proliferation and Induces Apoptosis of Pathological Scar Fibroblasts Through the Mechanism Involving TGF-β1/Smads Signaling Pathway[J].Cell biochemistry and biophysics,2015,71(3):1267-1272.DOI:10.1007/s12013-014-0317-6

[5] Lin CY,Hsieh YH,Yang SF,et al.Cinnamomum cassia extracts reverses TGF-β1-induced epithelial-mesenchymal transition in human lung adenocarcinoma cells and suppresses tumor growth in vivo[J].Environmental toxicology,2017,32(7):1878-1887.DOI:10.1002/tox.22410

[6] Hyeonseok Ko.Geraniin inhibits TGF-β1-induced epithelial-mesenchymal transition and suppresses A549 lung cancer migration, invasion and anoikis resistance[J].Bioorganic & medicinal chemistry letters,2015,25(17):3529-3534.DOI:10.1016/j.bmcl.2015.06.093

[7] Xu Y,Lou Z.Arctigenin represses TGF-β-induced epithelial mesenchymal transition in human lung cancer cells[J].Biochemical and biophysical research communications,2017,493(2):934-939.DOI:10.1016/j.bbrc.2017.09.117

[8] Lim WC,Kim H,Kim YJ,et al.Dioscin suppresses TGF-β1-induced epithelial-mesenchymal transition and suppresses A549 lung cancer migration and invasion[J].Bioorganic & medicinal chemistry letters,2017,27(15):3342-3348.DOI:10.1016/j.bmcl.2017.06.014 

Point 5:Pathophysiological relevance of TGF, inhibitors, and Vicenin II doses examined should be extensively explained in the Discussion.

Response 5:Thank you for your valuable suggestion.We have already explained the doses choice and pathophysiological relevance of TGF-β1 and Vicenin II in the Discussion (lines 274-280).In addition,we have added pathophysiological relevance of inhibitors LY294002 and SB431542 in Discussion (lines 298-306).

Point 6:Grammatical errors should be professionally corrected.

Response 6:Thank you for your advise.We have already sent the manuscript to MDPI Publications (https://www.mdpi.com/authors/english) in November 23,2018 before we submit to “Molecules” and the ID was english-6735.If it has any other grammatical errors, please point out the positions in the manuscripts and contact us to revise it.

Reviewer 2 Report

The article by Wei and co-workers describes the ability of vicenin II to inhibit TGF-induced epithelial-mesenchymal transition via the deactivation of TGF/smad and PI3K/Akt/mTOR signaling. Inhibition of EMT may prevent migration and invasion of tumor cells. The compound vicenin II is a glycosidated flavonoid which has previously been found to exhibit anti-cancer activity. Since many of the findings have already been reported in the literature, the article lacks novelty. However, the data reported were good and overall the article does a nice job of putting previously known facts together, supported by experimental data, in a well presented and streamlined fashion. However, there are several items that need to be addressed prior to its publication.

1.       Key references are missing. Some related published studies were referenced (Ref. 52), but many critical reports that already described the current findings were not mentioned. In fact, the commercially available (Sigma Aldrich) compound has previously been reported to inhibit TGF induces signaling (Lee W. Inflammation 2015) as well as PI3K/Akt/mTOR signaling (Buruah, BioFactors, 2018).

2.        Since many of the results were previously published the article loses novelty and its claims of novelty should be deleted. Similarly, the need and emphasis of the structure identification of the compound (in title, results and experimental) is highly questionable since this well-known compound is even commercially available and previous syntheses are published. If authenticity is determined by comparison to a previous reference, those reference(s) should be cited.

3.       Vincenin II also inhibits other pathways (for example NF-kB), but he authors do not evaluate to what extend these are responsible for some of the observed results. Are the effects related to EMT exclusively related to TGF signaling or due to a broader effect on multiple pathways?

4.       There is a lot of data reported on the concentrations of vicenin II required to elicit a physiological response in vivo (animal models). The authors should correlate those studies with the current findings to determine whether the current cellular findings could be responsible for observed physiological responses seen in vivo.

Author Response

The article by Wei and co-workers describes the ability of vicenin II to inhibit TGF-induced epithelial-mesenchymal transition via the deactivation of TGF/smad and PI3K/Akt/mTOR signaling. Inhibition of EMT may prevent migration and invasion of tumor cells. The compound vicenin II is a glycosidated flavonoid which has previously been found to exhibit anti-cancer activity. Since many of the findings have already been reported in the literature, the article lacks novelty. However, the data reported were good and overall the article does a nice job of putting previously known facts together, supported by experimental data, in a well presented and streamlined fashion. However, there are several items that need to be addressed prior to its publication.

Point 1: Key references are missing. Some related published studies were referenced (Ref. 52), but many critical reports that already described the current findings were not mentioned. In fact, the commercially available (Sigma Aldrich) compound has previously been reported to inhibit TGF induces signaling (Lee W. Inflammation 2015) as well as PI3K/Akt/mTOR signaling (Buruah, BioFactors, 2018).

Response 1:Thank you for your nice comments.We have replenished this studies in the introduction (lines 95-98).

Point 2:Since many of the results were previously published the article loses novelty and its claims of novelty should be deleted. Similarly, the need and emphasis of the structure identification of the compound (in title, results and experimental) is highly questionable since this well-known compound is even commercially available and previous syntheses are published. If authenticity is determined by comparison to a previous reference, those reference(s) should be cited.

Response 2:Thank you for your comments.The studies on Dendrobium officinale have mainly focused on the polysaccharides, but flavonoids are a group of phytochemicals with diverse biological functions and significant substances in Dendrobium officinale that should not be ignore.Thus, we extract and isolate many flavonoids from Dendrobium officinale and then identify their structure.Vicenin Ⅱ is one of them.We didn't know the commercial availability of Vicenin Ⅱ until we identify its structure.We have added the comparative reference in the results (lines 132-134).

Point 3:Vincenin II also inhibits other pathways (for example NF-kB), but he authors do not evaluate to what extend these are responsible for some of the observed results. Are the effects related to EMT exclusively related to TGF signaling or due to a broader effect on multiple pathways?

Response 3:Thank you for your comments.Most studies of Vincenin II inhibits other pathways (for example NF-kB) was about anti-inflammatory while the anti-metastatic effects of Vicenin Ⅱ have not been investigated.In this study, we investigated the effect of key protein expression of TGF-β/Smad and PI3K/Akt/mTOR signaling pathways. In order to recomfirm whether Vincenin II works via TGF-β/Smad and PI3K/Akt/mTOR signaling pathways, we added pathway inhibitors SB431542 (TGF-β/Smad) and LY294002 (PI3K/Akt/mTOR) as a reference group.These results is enough to demonstrate Vincenin II inhibits TGF-β/Smad and PI3K/Akt/mTOR signaling pathways. and we think is not necessary to evaluate other pathways.

On the other hand, we use TGF-β1 to induce metastasis in this study and we mainly focus on the effect on TGF-β/Smad and PI3K/Akt/mTOR signaling pathways.So we think is not necessary to evaluate a broader effect on multiple pathways.

Point 4:There is a lot of data reported on the concentrations of vicenin II required to elicit a physiological response in vivo (animal models). The authors should correlate those studies with the current findings to determine whether the current cellular findings could be responsible for observed physiological responses seen in vivo.

Response 4:Thank you for your valuable suggestion.We would have considered discuss those studies, but the reports of vicenin II on the physiological response in vivo almost was the anti-inflammatory activity and the animal models was inflammation models or spasmodic models such as different mediums induced vascular inflammatory responses(Lee IC. Inflammation research,  2015),(Ku SK.Canadian journal of physiology and pharmacology, 2016),(Kang H.Inflammation research, 2015) and anti-spasmodic effect in rat ileum(Eugen J.Phytomedicine, 2013), et al.It is not belong to the same field as epithelial-mesenchymal transition and metastayic resistance.After much consideration, we think those studies don’t have much reference value so that we put emphasis on the studies related to EMT in whole discussion since the anti-metastatic effects of Vicenin Ⅱ have not been investigated previously.
